# Harnessing Bacterial Endophytes for Promotion of Plant Growth and Biotechnological Applications: An Overview

**DOI:** 10.3390/plants10050935

**Published:** 2021-05-07

**Authors:** Ahmed M. Eid, Amr Fouda, Mohamed Ali Abdel-Rahman, Salem S. Salem, Albaraa Elsaied, Ralf Oelmüller, Mohamed Hijri, Arnab Bhowmik, Amr Elkelish, Saad El-Din Hassan

**Affiliations:** 1Department of Botany and Microbiology, Faculty of Science, Al-Azhar University, Nasr City, Cairo 11884, Egypt; aeidmicrobiology@azhar.edu.eg (A.M.E.); mohamedali@kyudai.jp (M.A.A.-R.); salemsalahsalem@azhar.edu.eg (S.S.S.); albraa.mahmoud@azhar.edu.eg (A.E.); 2Department of Plant Physiology, Matthias Schleiden Institute of Genetics, Bioinformatics and Molecular Botany, Friedrich-Schiller-University, 07743 Jena, Germany; b7oera@uni-jena.de (R.O.); amr.elkelish@science.suez.edu.eg (A.E.); 3Biodiversity Centre, Institut de Recherche en Biologie Végétale, Université de Montréal and Jardin botanique de Montréal, Montréal, QC 22001, Canada; mohamed.hijri@umontreal.ca; 4African Genome Center, Mohammed VI Polytechnic University (UM6P), 43150 Ben Guerir, Morocco; 5Department of Natural Resources and Environmental Design, North Carolina A&T State University, Greensboro, NC 27411, USA; abhowmik@ncat.edu; 6Botany Department, Faculty of Science, Suez Canal University, Ismailia 41522, Egypt

**Keywords:** biotechnological applications, endophytes, plant growth, biofertilizers, phytohormones, phytoremediation

## Abstract

Endophytic bacteria colonize plants and live inside them for part of or throughout their life without causing any harm or disease to their hosts. The symbiotic relationship improves the physiology, fitness, and metabolite profile of the plants, while the plants provide food and shelter for the bacteria. The bacteria-induced alterations of the plants offer many possibilities for biotechnological, medicinal, and agricultural applications. The endophytes promote plant growth and fitness through the production of phytohormones or biofertilizers, or by alleviating abiotic and biotic stress tolerance. Strengthening of the plant immune system and suppression of disease are associated with the production of novel antibiotics, secondary metabolites, siderophores, and fertilizers such as nitrogenous or other industrially interesting chemical compounds. Endophytic bacteria can be used for phytoremediation of environmental pollutants or the control of fungal diseases by the production of lytic enzymes such as chitinases and cellulases, and their huge host range allows a broad spectrum of applications to agriculturally and pharmaceutically interesting plant species. More recently, endophytic bacteria have also been used to produce nanoparticles for medical and industrial applications. This review highlights the biotechnological possibilities for bacterial endophyte applications and proposes future goals for their application.

## 1. Introduction

Endophytes are fungal and bacterial organisms that inhabit the plant endosphere without harming their hosts. They live asymptomatically in the plant cellular environment and perform symbiosis-specific functions such as synthesis of secondary metabolites or signaling molecules that function as internal and external stimuli during the mutualistic interaction [1]. Endophytic microbes are sources of novel biomolecules for the biochemical and pharmaceutical industries [2]. They produce biologically active metabolites, including immune-suppressive compounds, anticancer agents, plant growth promotors, antimicrobial volatiles, insecticides, antioxidants, and antibiotics [3,4], with huge potential for application in medicine, pharmaceutics industry, or agriculture (Figure 1). Moreover, endophytic microbes can improve plant growth under harsh conditions such as nutrient stress, temperature stress, salinity, trace metal stress, or drought [5]. They can also help plants to grow in contaminated environments by degrading hazardous compounds. We describe the main concepts of the application of endophytes in agriculture and biotechnology.

## 2. Types of Bacterial Endophytes

Taxonomically, endophytic bacteria belong to 16 phyla comprising more than 200 genera. However, the majority of them belong to the three phyla: Firmicutes, Actinobacteria, and Proteobacteria [6]. They are either Gram-negative or Gram-positive, such as *Pseudomonas*, *Achromobacter*, *Agrobacterium*, *Xanthomonas*, *Acinetobacter*, *Microbacterium*, *Bacillus*, and *Brevibacterium* [7]. Among others, mycoplasma has been isolated from the cytoplasm of marine green algae (e.g., *Bryopsis hypnoides* or *Bryopsis pennata*) [8].

Endophytic colonization can be local or systemic [9,10]. The plant endosphere represents a protective niche in which the endophytes are protected from biotic and abiotic stress. Moreover, endophytes can ecologically adapt to their environment and overcome plant defense responses [11].

Bacterial endophytes can also be classified as obligate or facultative endophytes. When endophytic bacteria rely on plant metabolites for survival and transfer between plants vertically or through the activity of different vectors, they are defined as obligate endophytes [12]. In comparison, facultative endophytes live outside the host at a definite stage of their life and are usually transmitted to plants from the surrounding atmosphere and soil [13]. The data presented in Table 1 show some examples of culturable bacterial endophytes and their attributes as plant growth-promoters.

## 3. Plant–Bacterial Endophyte Interactions

Interactions between bacteria and plants occur in many ways and at different levels (Figure 2). All plant organs interact with microorganisms at a specific stage of their life, and these interactions are not necessarily harmful to the plant. Plants can also benefit directly or indirectly from the interaction [37,38]. An example includes the well-studied rhizobia–legume interaction. Many endophytic bacteria form less specific symbiotic interactions with plants, although both partners adjust their metabolisms to the symbiotic conditions and can influence the biochemical properties of the partner [39]. This can result in promotion of the growth of the plant under normal and particularly harsh conditions [40,41].

Plants produce root exudates to attract beneficial bacteria, whereas bacterial endophytes recognize these compounds [42]. The bacteria in the root environment move towards the roots in response to the chemical attraction of the exudates [43]. After attachment to the root surface, they can enter the root, e.g., at lateral root emergence or openings caused by wounds or mechanical wounding [43]. Several bacterial structures are involved in their attachment to the plant surface, including fimbriae, flagella, bacterial surface polysaccharides, and lipopolysaccharides. For example, in *Rhizobium* spp., the surface polysaccharides are modified during the transition from free-living cells to the bacteroid form through the expression of surface antigens [44]. 

Bacterial entrance and spread inside plant tissues require plant cell wall modification, which is achieved by the secretion of cell-wall-degrading enzymes like endoglucanases, xylanases, cellulases, and pectinases by the endophytic bacteria [45,46]. Penetration into the host can be active or passive. Active penetration is completed by proliferation and attachment tools involving pili, flagella, twitching motility, and lipopolysaccharides, whereas quorum sensing influences bacterial colonization and movement in the host plant [47]. Passive penetration occurs through cracks on root tips or root emergence zones or arising from the activities of harmful organisms [12].

Bacterial colonization often begins at the root surface. After successful entry, the bacteria can move to aerial parts by the transpiration stream and with the support of the bacterial flagella [48]. This colonization pattern begins with intracellular microbial access through root hairs [49]. Endophytes can pass through the plant cell wall and enter the root cell either directly by the secretion of plant cell-wall-degrading enzymes and passage through the plant plasma membrane or by rhizophagy. Rhizophagy is a phenomenon in which many plants get microbes from the soil into their cells and digest them as a source of essential nutrients [50,51]. 

However, most endophytes reside in the intercellular spaces of their hosts, i.e., in sites that are rich in carbohydrates, inorganic nutrients, and amino acids [52]. Besides root colonization, endophytic bacteria can also occupy the intercellular spaces of stems, leaves, seeds, flowers, fruits, and xylem vessels [53,54,55,56,57]. Endophytes with intracellular colonization are difficult to study because they are often non-cultivable [58].

## 4. Applications of Bacterial Endophytes 

### 4.1. Agricultural Applications

#### 4.1.1. Plant Growth Promotion

Endophytic bacteria utilize the same mechanisms as rhizosphere bacteria for enhancing plant growth in silviculture, horticulture, and agriculture, as well as in phytoremediation [59]. The growth-promoting effect can be either direct or indirect (Figure 3).

##### Direct Plant Growth Promotion

A.Phytohormone production
Plant growth regulators or phytohormones are organic substances that modify, inhibit, or promote plant growth and development at low concentrations (<1 mM) [60]. For agricultural applications, they are either chemically synthesized, extracted from plant materials, or produced by microbial fermentation. However, their complex chemical structures or low abundance in plants often prevent the application of these techniques [61]. Alternatively, the development of techniques for fermentative hormone production may decrease the production costs and increase productivity.

The main phytohormones produced by endophytic bacteria are auxins, cytokinins, abscisic acid, ethylene, brassinosteroids, gibberellins, strigolactones, and jasmonates [59,62,63,64]. 

Indole-3-acetic acid (IAA) primarily promotes plant cell elongation and differentiation [65]. In symbiosis, the IAA produced by the bacteria stimulates adventitious and lateral root development, promotes nutrient access, and enhances root exudation to increase the interaction [66,67]. Common endophytic IAA producers belong to the bacterial genera *Azospirillum*, *Azotobacter*, *Alcaligenes*, *Herbaspirillum*, *Enterobacter*, *Pseudomonas*, *Klebsiella*, *Rhizobium*, *Burkholderia*, *Pantoea*, *Bacillus*, *Acetobacter*, and *Rhodococcus* [68,69,70,71]. They are often found in nature, as recently shown for medicinal plant populations in Egypt [36,72]. 

The plant defense hormone ethylene is involved in various stress responses, as well as developmental processes, including root development. In the rhizobia–legume symbiosis, it participates in the nodulation process [73]. Bacterial endophytes can possess a 1-aminocyclopropane-1-carboxylate deaminase (ACC deaminase), which generates the nitrogen sources ammonia and α-ketobutyrate from the ethylene precursor ACC [74]. Therefore, these bacterial endophytes can promote plant growth under nitrogen-limitation conditions through the secretion of ACC deaminases. This also results in a stronger immune system and promotes tolerance against abiotic stress. For instance, *Pseudomonas brassicacearum* SVB6R1 increases salt stress tolerance in sorghum plants through the section of ACC deaminase [33]. Moreover, *Paenibacillus polymyxa* from the bulbs of *Lilium lancifolium* promotes the growth of two *Lilium* varieties through secretion of ACC deaminase, among other effects [34]. 

Several studies reported endophytic bacterial species which produce gibberellins and cytokinins. Gibberellins control plant growth and development through enhanced seed germination, stem and leaf growth promotion, stimulation of flowering and fruit development, and delaying plant aging [75,76]. Cytokinin controls cell division and differentiation, increases resistance to biotic and abiotic stress, enhances phloem transport, and promotes flowering and axillary bud growth [77]. The endophytic *Azospirillum lipoferum* was inoculated in a maize plant previously treated by gibberellin inhibitor synthesis and subjected to drought stress. The inoculated maize plant performed better than the uninoculated controls, particularly under stress due to the bacterial gibberellin [78]. Moreover, endophytic bacterial strains including *Pseudomonas resinovorans* and *Paenibacillus polymaxa* isolated from *Gynura procumbens* exhibited an efficient ability to produce compounds similar to cytokinin in a broth extract [79]. These examples of endophytic bacteria-mediated biosynthesis of different phytohormones are illustrated in Table 2.
B.Biofertilization
a.Nitrogen fixation

Manufacturing nitrogenous fertilizers is costly, and their usage increases the pollution of groundwater with nitrate [90]. Interestingly, biological nitrogen fixation (BNF) accounts for about two-thirds of the globally fixed nitrogen [91]. Apart from photosynthesis, biological nitrogen fixation (BNF) is the most remarkable biological process, and this process is limited to prokaryotic organisms only. Bacterial endophytes can establish closed relations with certain crops. The plant endosphere contains excess carbon and a lack of oxygen, which presents suitable conditions for nitrogen fixation that can be then transported by endophytes to their host plant. Interestingly, endophytic bacteria can fix nitrogen within plants without forming nodule-like structures [92]. Some endophytic bacteria possess BNF genes that enable them to convert nitrogen gas (N_2_) to other nitrogen forms such as nitrate and ammonium, which can be utilized by host plants [93,94]. In Brazil, it was observed that sugarcane plant (*Saccharum officinarum* L) cultivated without nitrogen fertilizers and infected with diazotrophic endophytes such as *Herbaspirillum seropedicae* and *Acetobacter diazotrophicus* were able to derive all their nitrogenous needs from atmospheric N_2_ [95]. Regarding nitrogen fixation, endophytes do better than rhizosphere microbes in promoting plant growth and health, and help the plant thrive in nitrogen-restricted soil [96]. 

Bacterial endophyte inoculation has contributed to increased biomass over un-inoculated control plants, as noted by several authors (Table 3). Some endophytic strains of *Enterobacter* spp. and *Klebsiella* spp. have been reported to fix N_2_ and promote the growth of sugarcane under gnotobiotic and natural conditions [97,98,99,100]. Wei et al. [101] isolated *Klebsiella variicola* DX120E, a nitrogen-fixing bacterium, from the roots of the ROC22 sugarcane cultivar. This strain was capable of fixing N_2_ in association with sugarcane plants under gnotobiotic conditions, promoting GT21 cultivar growth and plant uptake of N, P, and K under greenhouse conditions. The total nitrogen content of *Poa pratensis* L. plants was increased by 37% after inoculation with the endophytic *Burkholderia vietnamiensis* WPB strain [102]. Significant increases in root and shoot biomass and flowers/fruits were obtained by the inoculation of cherry tomato *Lycopersicon lycopericum* cv ‘Glacier’, with *Rahnella* strain WP5, while an approximately twofold increase in total nitrogen content in root tissue of *Lolium perenne* was reported when the plant was inoculated with a multi-strain endophytic consortium (PTD1, WPB, WP19, WP1, and WW6) [103]. Andrade et al. [104] reported that about 40 endophytic bacterial strains were isolated from banana roots (*Musa* L.); out of these, 20 endophytic isolates had the capacity to grow in N-free media. Out of 20 isolates, four endophytic strains belonging to the Firmicutes (*Bacillus* spp.) showed the ability to fix nitrogen when assessed by the Kjeldahl and acetylene reduction assay methods. Moreover, these four isolates exhibit banana growth-promoting activities in vivo through nitrogen fixation, IAA production, and phosphate solubilization. The diazotrophic endophytic bacteria and their nitrogen-fixing capacity for plant growth promotion are summarized in Table 3.
b.Phosphate Solubilization

Plants take up monobasic (H_2_PO_4_^−^) and dibasic phosphate (HPO_4_^2−^) from the soil, whereas 95‒99% of soil phosphorus is non-available for plants because it is present in precipitated, immobilized, and insoluble forms [119]. In agriculture, phosphate deficiency is compensated by applying chemical or organic phosphate fertilizers [120]. 

Endophytes increase phosphorus availability for plants, as they dissolve insoluble phosphate via secretion of organic acids, chelation, and/or ion exchange [121]. Furthermore, endophytes can secrete P-solubilizing enzymes, such as phosphatase, phytase, and C—P lyase (Figure 4) [122]. Secretion of organic acids such as citric, malonic, fumaric, tartaric, gluconic, acetic, or glycolic acid is considered the main mechanism involved in P solubilization. The P-solubilizing activity of the bacteria is usually correlated with a decrease in the pH value of the medium, which varies due to bacterial species (Figure 4). For example, the phosphate-solubilizing *Bacillus* spp. secretes itaconic, lactic, isobutyric, isovaleric, and acetic acid, and thus acidifies the medium and rhizosphere during symbiosis [123]. Other P-solubilizing mechanisms are the secretion of exopolysaccharides (Figure 4). [124] and the degradation of P-containing substances.

Although it is not clear how bacteria living in a plant can dissolve the insoluble phosphate that is located outside of the plant, 50% of the endophytes isolated from ginseng were able to dissolve phosphate [125]. Comparably high numbers of endophytes isolated from soybean, cactus, legumes, strawberry, and sunflower were effective phosphate solubilizers [126,127,128]. Very high phosphate-solubilizing activity was recorded for the *Bacillus* species of *B. megaterium* and *B. amyloliquefasciens*, which solubilized insoluble zinc and potassium salts [129]. 

The utilization of endophytic bacteria inoculants as microbial biofertilizer candidates would provide a promising alternative to chemical fertilizers and for commercial applications. Inoculation of peanut (*Arachis hypogaea* L.) with the P-solubilizing nodule endophyte *Pantoea* J49 increased the aerial dry weight in greenhouse experiments [130]. *Pseudomonas letiola* mobilized insoluble P and increased the shoot length and growth of the apple tree variety Ligol [131]. Moreover, inoculation of *Rhodococcus* sp. EC35, *Pseudomonas* sp. EAV, and *Arthrobacter nicotinovorans* EAPAA enhanced the P-availability and plant growth of *Zea mays* in soils amended with tricalcium phosphate [132]. Oteino et al. [133] found that the endophytic *Pseudomonas strains* L111, L228, and L321 isolated from the leaves of *Miscanthus giganteus* showed high P solubilization activity. *Pseudomonas fluorescens* L321 inoculation resulted in gluconic acid accumulation in the medium and consequently larger *P. sativum* L. plants. Joe et al. [134] investigated the effect of two salt-tolerant and phosphate-solubilizing bacteria (*Acinetobacter* sp. and *Bacillus* sp.) on *Phyllanthus amarus* and showed that the endophytes were responsible for the better performance and faster growth of the plants due to the promotion of germination, better P uptake, and stimulation of the immune system by promoting the biosynthesis of phenolic compounds, the radical scavenging system, and antioxidative enzymes.

Chen et al. [135] showed that the endophytic *Pantoea dispersa* promoted P uptake into cassava (*Manihot esculenta* Crantz) roots due to the secretion of salicylic and benzene acetic acids for more effective P solubilization. Root inoculation with *P. dispersa* also activated the natural soil microbial community in the rhizosphere. Such strains could be suitable for optimizing agro-microecological systems under P limitations. Finally, Castro et al. [136] used the mangrove endophytic *Enterobacter* sp. as an inoculum for the production of seedlings of *A. polyphylla* trees and found that the endophyte increased shoot dry mass and the fitness of the seedling.
c.Siderophore Production

Iron is essential for all life and is required as a co-factor of many essential enzymes, including the biochemical processes involved in nitrogen fixation in nodules. However, most soil iron is not available for plant absorption because it is present in extremely insoluble ferric (Fe^3+^) forms of carbonates, hydroxides, oxides, and phosphates [137] in the soil. Under iron deficiency, many microorganisms can produce and secrete the low molecular weight siderophores, which bind Fe^3+^, Fe^2+^, and other divalent metal ions which are essential for plants [138]. Besides delivery to the plants, siderophores also participate in scavenging undesired metal ions in the rhizosphere to prevent uptake by the roots. These include Zn, Cd, Cr, Al, and Pb ions and, also radioactive ions such as U or Np to reduce toxicity for the plants [139]. Siderophores can also stimulate the plant-induced systematic resistant response to alleviate the toxicity of trace metals to plant growth [76]. 

Normally, endophytic bacteria produced siderophores only in soils with iron limitations [140]. Sabaté et al. [83] showed this for siderophores produced by two endophytic *Bacillus* spp. strains, which, in promoted the growth of common bean under iron limitation. Siderophores of the bacterial endophytes also outcompete phytopathogen by binding the essential elements required for their propagation, thereby protecting the plants [141]. For instance, endophytic *Bacillus* spp. inhibit the growth of *F**usarium oxysporum* by absorbing Fe^3+^ from the environment through the secretion of siderophores. Lacava et al. [142] showed that the citrus endophyte *Methylobacterium mesophilicum* released hydroxamate-type siderophores into the medium. When the siderophore-containing supernatant was applied to *Xylella fastidiosa* subsp. *pauca*, it promoted the growth and chlorophyll content of the plants. Similar results were reported for various symbiotic interactions with different agriculturally relevant hosts in greenhouse experiments.

##### Indirect Plant Growth Promotion

A.Stress Tolerance
Plants are exposed to various stresses [143], which are the result of antagonistic or toxic substances in their environment, nutrient or essential ion limitations, or biotic stress inducers [144]. Both biotic and abiotic stresses contribute about 30–50% of the agricultural loss worldwide [145]. Temperature, salinity, drought, trace metals, flooding, and nutrient deficiency are considered the major abiotic stresses (Figure 5). Biotic stresses are induced by pathogenic microorganisms up to insects or nematodes. Therefore, future studies aim to develop eco-friendly technologies which increase plants’ resistance to both abiotic and biotic stresses. A promising strategy is the generation of stronger and healthier plants with an improved immune system. Not surprisingly, endophytes have long been known to fulfill these criteria and provide excellent systems for agricultural application, as well as the identification of the molecular mechanisms by which they promote plant fitness. The microbial strategies range from the immediate activation of responsive systems which directly and specifically counteract the stress [146] and the production of anti-stress metabolites [147] that strengthens the plants, to a broad spectrum of immune responses which ultimately protect the plants better when exposed to different stresses. Understanding the mechanism behind these strategies may provide us with molecular and biochemical tools for agricultural applications.

Endophytic bacteria support plants to combat drought stress via the production of volatile compounds, abscisic acid, ACC-deaminase, and IAA. Moreover, enhanced antioxidant activity and osmotic adjustment participate in the endophyte effects [148]. Razzaghi-Komaresofla et al. [149] introduced salinity-tolerant *Staphylococcus* sp. strains which improved the growth and salt tolerance of *Salicornia* sp. plants either individually or in combination. Endophyte-managed host resistance to pathogens also includes niche competition by the production of defensive metabolites such as antibiotics, antimicrobial, and structural compounds, as well as the induction of immunity or systemic resistance in the host plant [150].

Plants experiencing biotic and abiotic stress produce elevated concentrations of ethylene [151], which is are synthesized from ACC [152]. Production of the defense hormone ethylene restricts growth and hence affects the development of the plant in general [153]. Afridi et al. [154] reported the improved growth and stress tolerance of two wheat varieties inoculated with ACC-producing bacterial endophytes. An advantage of ethylene-producing endophytes may be the local synthesis of the hormone in the symbiotic tissue, which might ensure that ethylene-induced defense occurs only in tissues that are exposed to stress. Endophyte-produced ACC-deaminase can alleviate the impact of elevated ethylene concentrations on stressful plants through the hydrolysis of ACC into α-ketobutyrate and ammonia. The stressed plant could utilize the ammonia and energy liberated from ACC decomposition for growth [155,156]. Multiple combinations of the beneficial features of endophytes, such as combinations of ethylene, siderophore, or exopolysaccharide production; nitrogen fixation; and phosphate solubilization are important to strengthen the plants under stress or nutrient limitations [148,157,158]. Interestingly, some endophytic bacteria possess sigma factors (Table 4), which are used to change the expression of some genes under unfavorable conditions to reduce negative impacts [106]. Data represented in Table 4 showed some examples of stresses and the mechanisms utilized by bacterial endophytes to alleviate these stresses.
B.Endophyte-Based Phytoremediation 

Phytoremediation is a promising tool for cleaning soil, water and air of contaminants, and endophyte-assisted phytoremediation is used for metal bioremediation in the soil to allow or promote plant growth in previously contaminated soils [137]. Other investigations reported the use of endophytic remediation for contamination with organic contaminants [181], hydrocarbons [182], explosives [183], herbicides [184], tannery effluent [185], and uranium [186]. 

Eevers et al. [187] examined the effect of *Enterobacter aerogenes* UH1, *Sphingomonas taxi* UH1, and *Methylobacterium radiotolerans* UH1 on the growth of zucchini plants on 2,2-bis(p-chlorophenyl)-1,1-dichloro- ethylene (DDE)-contaminated fields. Plants inoculated with the three strains separately or in combination showed an increased weight. Mitter et al. [188] planted sweet white clover plants on soils contaminated with diesel (up to 20 mg/kg). Plants inoculated with the hydrocarbon-degrading endophytes EA4-40 (*Pantoea* sp.), EA1-17 (*Stenotrophomonas* sp.), EA6-5 (*Pseudomonas* sp.), and EA2-30 (*Flavobacterium* sp.) showed increased biomass and successfully overcame the growth inhibition observed for non-inoculated plants. Wu et al. [189] isolated the endophytic *Bacillus safensis* strain ZY16 from the roots of *Chloris virgate*. This strain exhibited multifunctional properties, including efficient degradation of the C_12_–C_32_
*n-*alkanes of diesel oil and polycyclic aromatic hydrocarbons under hypersaline conditions, as well as production of biosurfactants, which resulted in stronger growth and biomass production in the inoculated plants compared with the controls. These studies indicated that bacterial endophytes have an important dynamic role in the management of abiotic stress and could efficiently be applied for environmental clean-up for sustainable agriculture development.
C.Disease Control

Currently, agrochemicals are considered the main method for combatting microbe-induced plant diseases; however, many of the applied substances have toxic effects on animals and humans. The application of endophytic bacteria is of great interest as an environmentally friendly alternative to agrochemicals [190]. 

Endophytes can restrict pathogen invasion into plants via direct and indirect mechanisms. The direct mechanism describes a competition between endophytes and phytopathogens in which the endophytes restrict pathogen growth, e.g., through the secretion of inhibitory metabolites. In the indirect mechanisms, the endophytes stimulate the plant’s immune system or increase the plant’s resistance toward the phytopathogens via upregulation of the defense genes [52]. Bacterial endophytes and phytopathogens have similar colonization patterns in the host plants and, consequently, they compete during invasion into host cells. Therefore, endophytes could be used as potential biocontrol agents to restrict pathogen entry into the host cell. The endophytic species *Pseudomonas fluorescens* and *Pseudomonas aeruginosa* produce 2, 4-diacetylphloroglucinol, penazine-1-carboxylic acid, pyoleutirin, pyrrolnitrin, or hydrogen cyanide, which suppress the growth of phytopathogenic fungi [191]. The most common endophytic bacterial species used in phytopathogen control are *Bacillus* species, since they produce lipopeptides, which hydrolyze fungal hyphal membranes [192,193]. The destabilization of the fungal membrane promotes nutrient leakage and hence reduces the virulence of fungi [191]. The indirect mechanisms include the stimulation of a jasmonate/ethylene-based defense response against necrotrophic microbes and a salicylic acid-based defense against biotrophic microorganisms [191,192].
D.Competition for Space and Nutrients

Niche competition means the competition of endophytes with deleterious pathogens for space and substrates. Blumenstein et al. [194] examined the competitive interaction between endophytes and the aggressive pathogen *Ophiostoma novoulmi*, which is the causative agent of the virulent Dutch elm disease. Based on the carbon utilization profiles, they showed that the endophytes showed extended niche overlap with the virulent pathogen; however, since the endophytes were more efficient at utilizing the applied carbon substrates, they were able to out-compete the pathogens. Another example described endophyte-produced siderophores that supplied sufficient iron to the endophytes but not the pathogen [195]. The chelated iron also became available to the host plant [52], which ultimately inhibited pathogen growth in the host plant by restricting mycelial growth and spore germination [196].
a.Antibiosis

Antibiosis is the synthesis and release of molecules that kill or inhibit the growth of the target phytopathogen [54], which results in plant disease control. The antibiotics and volatile organic compounds (VOCs) comprise ketones, alcohols, esters, terpenes, aldehydes, sulfur compounds, and lactones. VOCs, with their low molecular weights, can directly affect the growth of the phytopathogens at low concentrations, although the mechanisms are not well understood, since their perception systems are unknown [197,198]. Jasim et al. [199] reported the biosynthesis of iturin, surfactin, and fengycin by an endophyte *Bacillus* species from *Bacopa monnieri*. Besides toxic effects to pathogens, numerous endophytic bacterial VOCs control the symbiotic relationship in an extremely competitive environment for the host [200]. They might stimulate the propagation of the endophytes but not the pathogens.
b.Parasitism

Parasitism happens when one microbe feeds on another, e.g., a phytopathogen. This results in complete or partial lysis of its cellular structures. In particular, antagonistic bacteria feed on the fungal phytopathogen cell wall materials such as proteins, chitin, and glucans [201]. Bacteria form lipopolysaccharides and produce cell wall lytic enzymes for the infection of their hosts [202], and these cell wall lytic enzymes also cause hydrolysis of the cell wall of the fungal pathogen [203].
c.Induced Systemic Resistance (ISR)

Beneficial bacterial induce ISR in plants, which stimulates their local and systemic defensive responses, thus protecting the host against pathogen attacks. Besides the classical salicylic acid-based ISR, several endophytes also activate jasmonic acid- and ethylene-assisted immune responses [204]. An example of the latter mechanism includes the endophytic *Bacillus velezensis* YC7010, which confers systemic resistance in *Arabidopsis* seedlings against the insect pests *Myzus persicae* and the green peach aphid [205]. The different strategies are summarized in Table 5 and examples are provided in Figure 6. 

### 4.2. Biotechnological Applications

#### 4.2.1. Production of Bioactive Metabolites for Agricultural and Medical Applications

Endophytic bacteria produce diverse bioactive metabolites that can be used in medicine as well as in agriculture for plant growth promotion and pesticides. The following examples describe some of the metabolites with biotechnological relevance.

##### Pharmaceutical Applications

Newman and Cragg [217] argued that endophytes produced approximately half of the new drugs introduced in the market from 1981 to 2010. They are used as insecticides, antioxidants, antimicrobial agents, anticancer, and antidiabetic compounds, among others (Figure 7) [218]. Many metabolites used as anticancer or antimicrobial agents have multiple targets in plants, animals, and human pathogens and offer many prospects in veterinary and medical therapy. Several of them are also considered eco-safe [219].

Medicinal plants have long been used to cure many diseases. More recently, it became obvious that many of their important chemical ingredients were derived from endophytes or the association of the medicinal plants with them. Alvin et al. [220] isolated endophytes from medicinal plants producing polyketides and small peptides, which displayed antituberculosis activity. The secondary metabolite profiles produced by *Acinetobacter baumannii* associated with *Capsicum annuum* L. uncovered phenolic compounds with peroxidant and antioxidant abilities [221]. Many peptides with antibacterial, antifungal, anticancer, immunosuppressive, and antimalarial properties have been identified, and several of them are interesting because of their target-specificities [222]. 

#### 4.2.2. Industrial Applications

The extracellular hydrolytic enzymes produced by bacterial endophytes are required for host cell infection and triggering ISR [223]. The osmoregulation and antioxidant enzymes of the bacteria are involved in mitigating salinity stress on the metabolism of the host plant [224]. Microbial synthesis of biologically active compounds such as enzymes and secondary metabolites have been used for the manufacture of industrial products, including food and food supplements [225], biofuels [226], pharmaceuticals [227], detergents [228], and biopesticides [229]. 

For example, Ntabo et al. [230] isolated endophytic bacteria from Kenyan mangrove plants and showed that they produce proteases, pectinases, cellulases, amylases, and chitinases with putative industrial value. The endophytic *Pseudomonas aeruginosa* L10, associated with the roots of *Phragmites australis*, efficiently degraded hydrocarbons and produced a biosurfactant [231]. Baker et al. [232] studied endophytic bacteria associated with *Coffea arabica* L. and found that they could degrade caffeine, which has potential use in the decaffeination of beverages. 

#### 4.2.3. Nano Biotechnology

Nanoparticles have numerous technological and industrial applications in electronics, the energy industry, medicine, catalysis, and biotechnology [233,234,235,236,237,238]. Biological methods for the synthesis of nanoparticles provide many advantages over physical and chemical methods, avoiding the need for high energy and the absence of any toxic waste, which makes it simple, economical, and environmentally friendly [239,240,241]. 

Endophytic microbes reduce metallic ions for the production of nanoparticles [242] (Table 6). Low concentrations of nanosilver particles (AgNPs) synthesized by *Pantoea ananatis* showed antimicrobial activity against *Candida albicans* ATCC 10,231 and *Bacillus cereus* ATCC 10876, and higher concentrations were active against multidrug-resistant strains of *Enterococcus faecium* ATCC 700221, *Escherichia coli* NCTC 13351, *Streptococcus pneumoniae* ATCC 700677, and *Staphylococcus aureus* ATCC 33,592 [243]. AgNPs synthesized by endophytic *Streptomyces* spp. showed antimicrobial, antioxidant, and larvicidal activities [244]. The endophytic *Streptomyces* spp. isolated from medicinal plants synthesized Cu NPs and CuO NPs with antibacterial, antifungal, antioxidant, and insecticidal activities [245,246].

## 5. Conclusions and Future Prospective

Biotechnology has proven to be applicable in many medical, industrial, and agricultural fields, as it is inexpensive and eco-friendly. Endophytes are plant symbionts; unlike the rhizosphere and phyllosphere bacteria that live on the plant’s surface, endophytes find their way into the plants’ endosphere to become protected from inappropriate environmental conditions. Recently, different advanced techniques such as clustered regularly interspaced short palindromic repeats (CRISPR)/Cas-mediated genome editing (GE) have been used to explain the plant–microbe interaction. These advanced techniques develop ideal plants/microbes that are relevant for agricultural applications. Endophytes produce several active metabolites that have positive effects on plant growth, leading to increased crop yields. Compared with agrochemicals, the presence of endophytes inside the host plant puts them in direct contact with the plant and becomes more effective. Chemicals are applied outside the plant, and a large part of them do not benefit the plant and cause pollution of the environment. Apart from the biotechnological uses of endophytes in sustainable agriculture, bacterial metabolites include other compounds that can be used in many technological and industrial applications, as well as their ability to reduce metal ions to nanoparticles, which can be used in technological, industrial, medical, and electronic applications. In the future, nanoparticles will be helpful in the formation of new nano-drugs, nano-pestcides, and nano-fertilizers to suppress plant pathogens and to increase the fertility of soils and plants through providing essential elements. Therefore, the mechanisms of fabricating specific shapes and sizes using eco-friendly microbes, including endophytes, require more research. Moreover, the integration of bacterial endophytes into different biomedical and biotechnological applications is still limited; therefore, it is urgent to discover new compounds from endophytic bacteria that have biological activities. Much research is still needed to understand endophytic bacteria and their relationships with plants, and to optimize the use of their premium metabolic products and formulate them into products that can be marketed for economic benefit.

## Figures and Tables

**Figure 1 plants-10-00935-f001:**
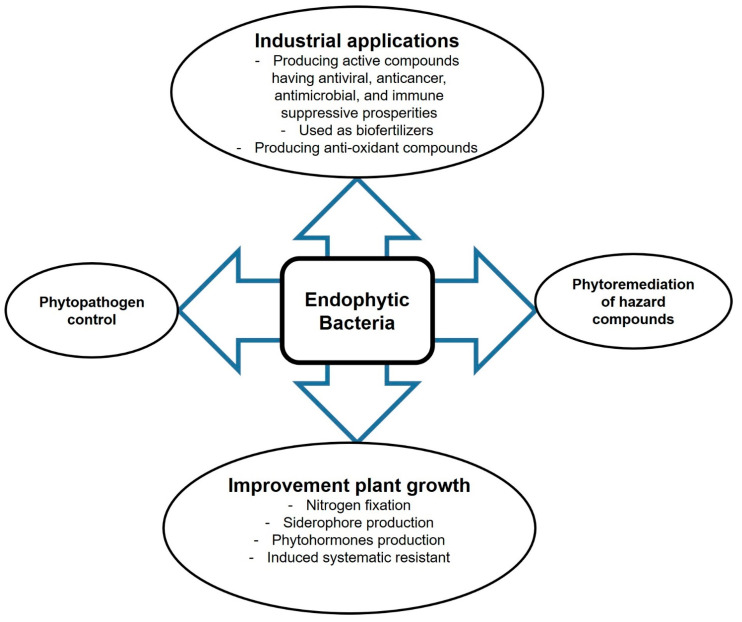
Prospective biotechnological applications of endophytic bacteria.

**Figure 2 plants-10-00935-f002:**
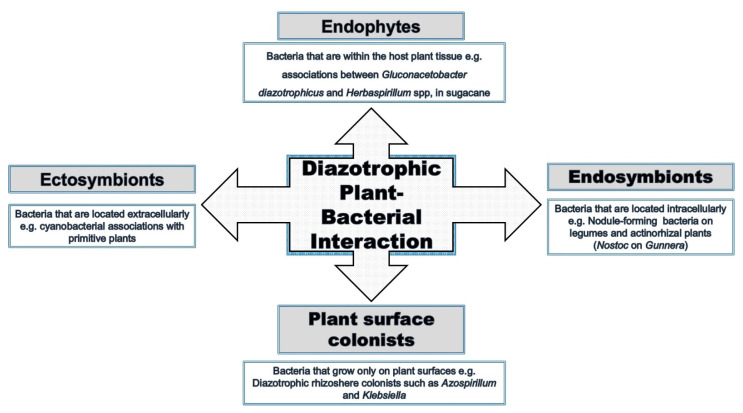
Endophytic plant–bacterial interactions versus other interactions.

**Figure 3 plants-10-00935-f003:**
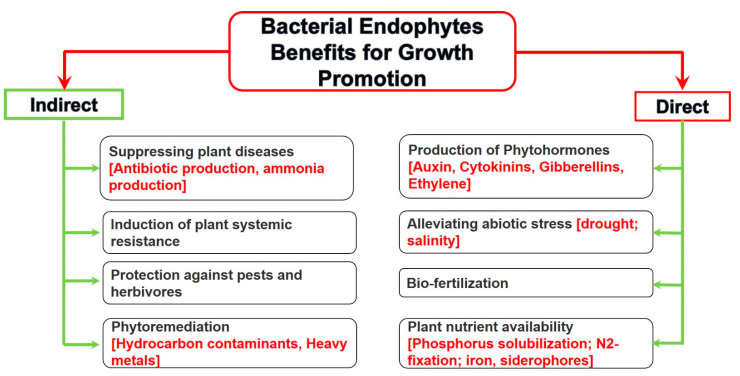
Direct and indirect growth-promoting attributes by endophytic bacteria.

**Figure 4 plants-10-00935-f004:**
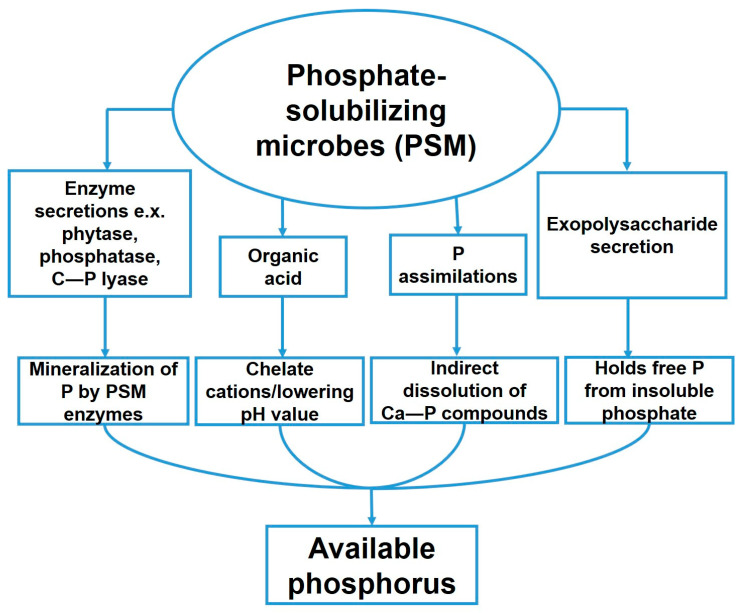
Different mechanisms utilized by phosphate-solubilizing microbes to solubilize phosphate.

**Figure 5 plants-10-00935-f005:**
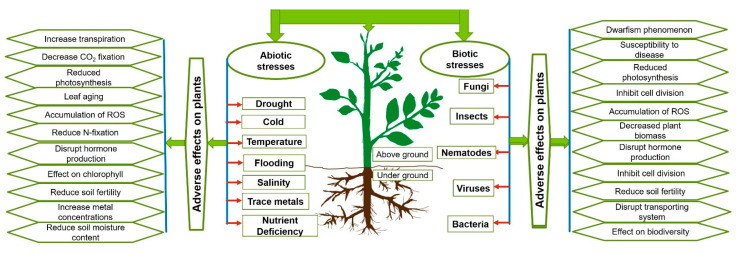
Biotic and abiotic stresses and their effects on plant growth.

**Figure 6 plants-10-00935-f006:**
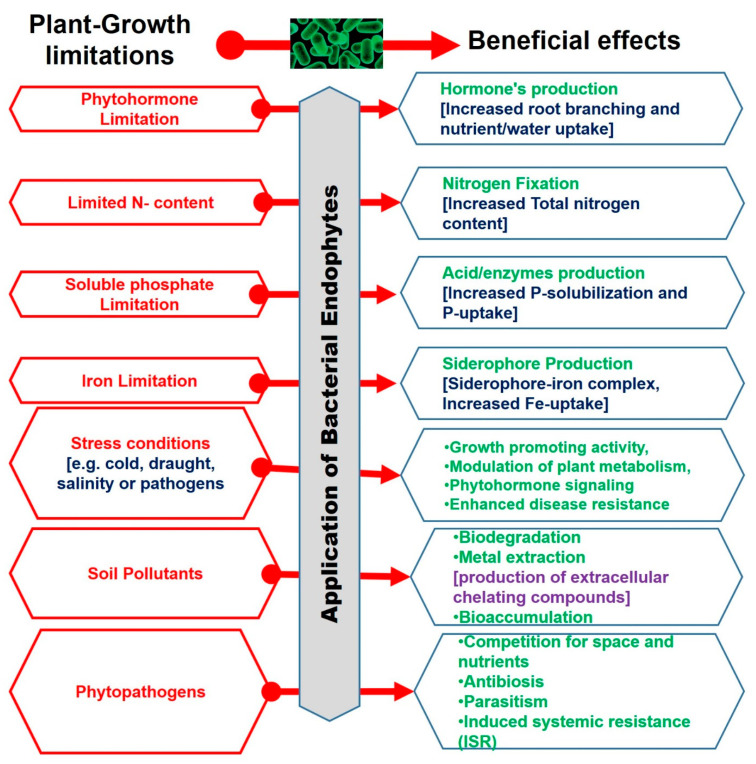
Challenges and benefits of bacterial endophyte utilization for applications in sustainable agriculture.

**Figure 7 plants-10-00935-f007:**
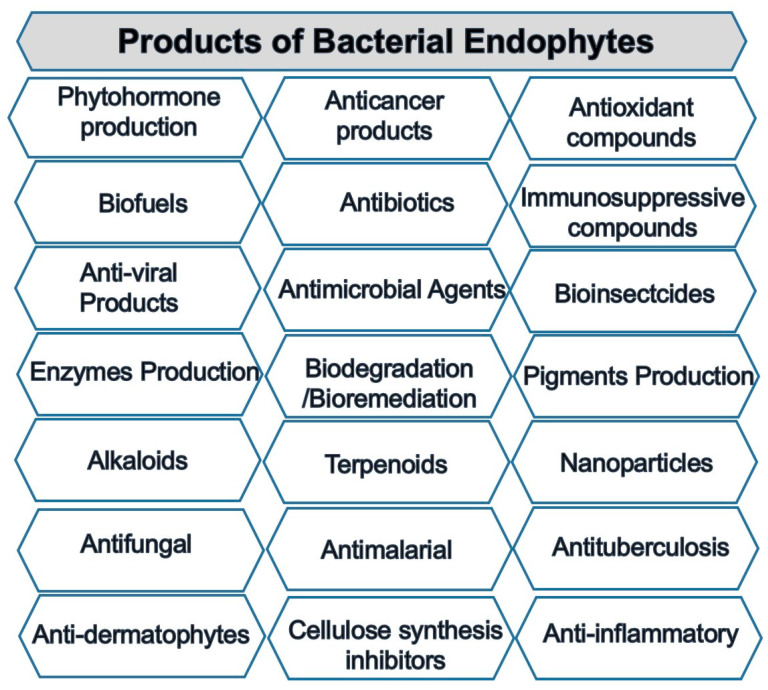
Diverse products produced by endophytic bacteria.

**Table 1 plants-10-00935-t001:** Some examples of recently reported culturable bacterial endophytes and their attributes as plant growth-promoters.

Endophytic Bacterial Species	Host Plant/ Organ	Plant Growth Promotion Attributes	References
**Proteobacteria**: *Pseudomonas* spp.	*Nicotiana tabacum*/seeds	Siderophores, IAA, ACC deaminase production, nitrogen fixation, phosphorus/potassium solubilization, and trace metal tolerance	[14]
**Firmicutes**: *Bacillus paralicheniformis*	Rice (*Oryza sativa* L.)/roots	Nitrogen fixation	[15]
**Firmicutes**: *Bacillus mojavensis*, *Bacillus* sp.	*Ammodendron bifolium*/roots and leaves	IAA, ACC deaminase, amylase, cellulase, protease, lipase production, phosphate solubilization, nitrogen fixation	[16]
**Proteobacteria**: *Aquabacterium*, *Duganella, Massilia, Bordetella, Salmonella, Pantoea, Kosakonia, Klebsiella, Serratia, Pseudomonas, Agrobacterium, Stenotrophomonas, Brevundimonas, Ancylobacter, Pleomorphomonas*. **Actinobacterium**: *Curtobacterium, Microbacterium, Nocardia, Sediminihabitans.* **Firmicutes**: *Bacillus, Micrococcus, Staphylococcus, Exiguobacterium*	*Sorghum bicolor*/roots and stems	IAA production, fungicidal and bactericidal activities, nitrogen fixation	[17]
**Proteobacteria**: *Acetobacter, Burkholderia, Caulobacter, Pseudomonas, Ralstonia, Bradyrhizobium, Methylocapsa*	*Pinus arizonica; Pinus durangensis*/roots, phloem, and bark	Production of active secondary metabolites, metabolism of vitamins and cofactors	[18]
**Actinobacteria**: *Streptomyces cavourensis*	*Cinnamomum cassia*/roots	Biosynthesis of active compounds with antimicrobial and cytotoxic properties and plant growth-promoting capabilities.	[19]
**Proteobacteria**: *Sphingomonas* sp.	*Tephrosia apollinea*/leaves	Drought tolerance	[20]
**Actinobacteria**: *Kocuria* sp., *Micrococcus luteus*	*Corchorus olitorius*/leaves, roots, seeds, and seedling	IAA and siderophore production.	[21]
**Firmicutes**: *Actinobacteria* **Proteobacteria**: *Curtobacterium* sp., *Microbacterium* sp., *Methylobacterium* sp., *Bacillus amyloliquefaciens*	Browntop millet/seeds	Auxin production, phosphate solubilization, inhibiting fungal pathogens	[22]
**Proteobacteria**: *Enterobacter ludwigii*, *Enterobacter* spp., *Agrobacterium tumefaciens*, *Kosakonia cowardii*, *Variovorax* sp., *Burkholderia* spp., *Pantoea vagans*, *Serratia marcescens* **Firmicutes**: *Bacillus* sp.	Soybean/roots, stems, and leaves	Antagonistic activity against soybean pathogenic fungi and bacteria	[23]
*Chryseobacterium endophyticum*, *Paenibacillus castaneae*, *Streptomyces* sp., *Lactobacillus plantarum*, *Bacillus proteolyticus*, *Pseudomonas* sp., *Serratia rubidaea*, *Klebsiella aerogenes*, *Paraburkholderia* sp., *Burkholderia* sp., *Bacillus cereus*, *Bacillus subtilis, Enterobacter cloacae*, *Enterobacter* sp., *Arthrobacter* sp., *Bacillus thuringiensis*, *Bacillus* sp.	Pigeonpea/stems, roots, and leaves	Antimicrobial activity against *Fusarium* wilt (*Fusarium udum*)	[24]
**Actinobacteria**: *Micrococcus yunnanensis*	*Avicennia marina*/Propagule teguments	IAA, ammonium, siderophore, and protease production	[25]
59 bacterial isolates belonging to phyla: **Proteobacteria**, **Firmicutes**, and **Actinobacteria**	Chickpea (*Cicer arietinum* L.)/roots	IAA production, ammonia production, cellulase production, salt tolerance	[26]
**Firmicutes**: *Bacillus velezensis*	Peanut/seeds	Antagonistic against *Sclerotium rolfsii*	[27]
**Firmicutes**: *Bacillus subtilis*	Sugarcane/leaves and stalks	Promoting plant growth, increasing N and chlorophyll content	[28]
**Proteobacteria**: *Delftia, Stenotrophomonas; Rhizobium; Brevundimonas, Variovorax;* *Achromobacter; Novosphingobium;* *Comamonas;* and *Collimonas*	*Zea mays* L., *Vicia faba* L., *Secale cereale* L., *Triticum aestivum* L., *Arctium lappa* L., and *Equisetum arvense* L./roots and stems	IAA and siderophore production, nitrogen fixation, and phosphate solubilization	[29]
**Proteobacteria**: *Enterobacter tabaci, Pantoea agglomerans, Stenotrophomonas maltophilia, Sphingomonas sanguinis, Enterobacter tabaci*	rice/seeds	IAA production and Cd tolerance	[30]
**Actinobacteria**: *Streptomyces niveus* NRRL 2466	*Camellia* spp. and related genera/roots and leaves	IAA, Ammonia, siderophores, ACC deaminase, chitinase, and protease, production. N_2_ fixation, P solubilization	[31]
138 endophytic bacterial strains belonging to the phyla Proteobacteria (*Pseudomonadales, Burkholderiales*, and *Xanthomonadales*) Firmicutes, and Bacteroidetes (*Bacillales* and *Flavobacteriales*)	Six terrestrial orchid species/roots	Phosphate solubilization, siderophore production, IAA production, antagonistic activities against plant pathogenic fungi	[32]
*Herbaspirillum lusitanum* (2 species), *Acinetobacter johnsonii* (3 species), *Stenotrophomonas rhizophila*, *Agrobacterium tumefaciens* (4 species), *Rhizobium radiobacter*, *Micrococcus yunnanensis*, *Paenibacillus graminis*, *Bacillus pumilus* (2 species), *Bacillus cereus; Bacillus muralis* (2 species), *Terribacillus goriensis*	Cucumber/roots, shoots, and leaves	IAA production, siderophore production, phosphate solubilization, antibiotic production, salt tolerance	[33]
*Bacillus cereus*, *Pseudomonas migulae* (3 species), *Pseudomonas* spp. (2 species), *Pseudomonas brassicacearum*, *Paenibacillus lautus*, *Brevibacterium frigoritolerans*, *Bacillus anthracis*, *Paenibacillus illinoisensis*, *Bacillus muralis*, *Bacillaceae bacterium*, *Micrococcus luteus*	Sorghum/roots	IAA production, siderophore production, phosphate solubilization, antibiotic production, salt tolerance	[33]
*Bacillus safensis*, *Acinetobacter lwoffii*, *Bacillus cereus* (6 species), *Bacillus thuringiensis* (4 species), *Bacillus muralis* (2 species), *Bacillus megaterium*, *Bacillus tequilensis*, *Bacillus aerophilus*, *Bacillaceae bacterium* (2 species), *Acinetobacter johnsonii* (2 species), *Microbacterium schleiferi*, *Bacillus subtilis*, *Paenibacillus* sp., *Bacillus niacin*, *Kochuria palustris*	Tomato/roots, shoots, and leaves	IAA production, siderophore production, phosphate solubilization, antibiotic production, salt tolerance	[33]
**Firmicutes**: *Paenibacillus polymyxa*	*Lilium lancifolium*/bulbs	IAA, siderophore, ACC deaminase, and organic acid production; nitrogen fixation; phosphate solubilization; antifungal activities against fungal phytopathogens	[34]
**Firmicutes** and **proteobacteria**: *Actinobacteria; Bacillus, Fictibacillus, Lysinibacillus, Paenibacillus, Cupriavidus,* and *Microbacterium*	Different rice cultivars such as Xiushui-48, Y-003, and CO-39/roots	Antagonistic effect against rice fungal phytopathogens	[35]
*Paenibacillus barengoltzii* (2 species), *Bacillus amyloliquefaciens* (2 species), *Bacillus thuringiensis* (2 species), *Bacillus cereus* (4 species)	*Fagonia mollis*/leaves	Enzymatic activities, IAA production, ammonia production, phosphate solubilization, antibiotic activities	[36]
*Brevibacillus agri* (3 species)	*Achillea fragrantissima*/leaves	Enzymatic activities, IAA production, ammonia production, phosphate solubilization, antibiotic activities	[36]

**Table 2 plants-10-00935-t002:** Some examples of endophytic bacterial strain-mediated biosynthesis of phytohormones.

Hormone	Producer Strain	Plant Source	Function/Effect	Reference
Gibberellins	*Bacillus amyloliquefaciens* strain RWL-1	*Oryza sativa* L. (Poales: Poaceae)	Plant growth promotion, hormone regulation	[62]
Abscisic acid	*Azospirillum lipoferum*	Maize	Alleviating drought stress symptoms in maize	[78]
Cytokinin’s	*Bacillus subtilis*	lettuce plants	Increased plant shoot and root weight by approximately 30%	[80]
Auxin (indol acetic acid)	*B. amyloliquefaciens B. cereus* and *Bacillus subtilis*	*Capsicum annuum* L. (Solanales: Solanaceae)	Anthracnose control, plant growth promotion, and biomass improvement	[81,82]
Auxins	*B. amyloliquefaciens* strain B14 and *Bacillus* sp. strains B19 and P12	*Phaseolus vulgaris*	Plant growth promotion, seed germination	[83]
Indol acetic acid	*B. subtilis* strain NA-108	*Fragaria ananassa* Duchesne (Rosales: Rosaceae)	Plant growth promotion and biomass improvement	[84]
IAA	*Pseudomonas aeruginosa.* *Bradyrhizobium* sp.	Soybean	Plant growth-promoting	[85]
IAA, gibberellins, and cytokinin	*Acitenobacter braumalli*, *Enterobacter asburiae*, *Pseudomonas aeruginosa*, *Pseudomonas fulva*, *Pseudomonas lini; Pseudomonas montelli*, *Pseudomonas putida*, *Pseudomonas thivervalensis*, *Sinorhizobium meliloti*, *Klebsiella pneumoniae*	Maize	Plant growth-promoting, alleviating drought stress, biocontrol activity	[86]
IAA	*Acinetobacter guillouiae*	Wheat	Plant growth-promoting;	[87]
IAA	*Arthrobacter sulfonivorans*	Wheat	Plant growth-promoting	[88]
IAA	*Acinetobacter calcoaceticus*, *Bacillus amyloliquefaciens*, *Enterobacter cloaca*, *Pseudomonas putida*	Soybean		[89]

**Table 3 plants-10-00935-t003:** Recent studies describing the diazotrophic endophytic bacteria, and their isolation and inoculation in agricultural plants.

Diazotrophic Endophytic Bacteria	Plant Source	Inoculated in	Capacity of N-Fixing Confirmed by	Reference
**Proteobacteria**: (*Acinetobacter calcoaceticus*, *Enterobacter cloacae, Pseudomonas putida*). Firmicutes: (*Bacillus cereus, Bacillus amyloliquefaciens*)	*Glycine max* L.	In vitro assay	Growth on Ashby’s N-free medium, *nif*H gene amplification	[89]
**Firmicutes**: (*Bacillus subtilis* EB-04, *Bacillus pumilus* EB-64, *Bacillus pumilus* EB-169, *Paenibacillus* sp. EB-144)	Banana tree	In vitro assay	Acetylene reduction assay *nif*H gene amplification	[104]
**Actinobacteria** (*Arthrobacter)*, **Proteobacteria** (*Rhizobium)*, **Firmicutes** (*Bacillus* spp.)	Diverse *Poaceae* family plants (maize, wheat, pearl millet, sorghum, and rice)	Wheat	Kjeldahl method	[105]
**Proteobacteria***(Pseudomonas aeruginosa* PM389)	*Pennisetum glaucum*	Wheat	Acetylene reduction assay *nif*H gene detection	[106]
**Proteobacteria** *(Herbaspirillum* sp.)	Tea plants (*Camellia sinensis* var. *assamic* and *C. sinensis*)	In vitro assay	Acetylene reduction assay *nif*H gene detection	[107]
**Proteobacteria** (*Burkholderia* spp., *Klebsiella* spp., *Novosphingobium* spp., *Sphingomonas* spp.)	Rice *(Oryza sativa)*	Rice *(Oryza sativa)*	Acetylene reduction assay	[108]
**Proteobacteria** (*Pseudomonas* spp., *Caballeronia sordidicola*, *Rhizobium herbae*) **Actinobacteria** (*Rathayibacter tanaceti*, *Frigoribacterium endophyticum*, *Herbiconiux solani*) **Bacteroidetes** (*Flavobacterium aquidurense*)	Lodgepole pine (*Pinus contorta* var. *latifolia*)	In vitro assay; lodgepole pine (*Pinus contorta*)	Acetylene reduction activity Amplification of *nif*H gene ^15^N isotope dilution assay	[109]
**Firmicutes***(Paenibacillus kribbensis* HS-R01, *Paenibacillus kribbensis* HS-R14)	Rice (*Oryza sativa* var. *japonica)*	Rice (*Oryza sativa* var. *japonica)*	*nif*H gene amplification	[110]
**Firmicutes** (*Bacillus* spp.) **Proteobacteria** *(Enterobacter* sp.)	*Zea mays* L.	*Zea mays* L.	Acetylene reduction assay *nif*H gene amplification	[111]
**Firmicutes** (*Paenibacillus polymyxa* P2b-2R)	Lodgepole pine (*Pinus contorta* var. *latifolia*)	*Zea mays* L.	Acetylene reduction assay *nif*H gene amplification ^15^N isotope dilution assay	[112]
**Firmicutes** (*Paenibacillus polymyxa* P2b-2R)	Lodgepole pine (*Pinus contorta* var. *latifolia*)	Canola (*Brassica napus* L.) and tomato (*Solanum lycopersicum*)	Acetylene reduction assay *nif*H gene amplification ^15^N isotope dilution assay	[113]
**Firmicutes**: (*Bacillus* spp, *Paenibacillus* spp.) **Proteobacteria**: (*Caballeronia* spp., *Pseudomonas* spp.)	Spruce tree	In vitro assay	Acetylene reduction assay	[114]
**Proteobacteria**: (*Burkholderia, Sphingomonas, Bradyrhizobium* sp., *Azospirillum brasilens*, *Rhodospirillum rubrum*, *Rhodobacter capsulatus*) **Cyanobacteria**: (*Nostoc punctiforme*) **Euryarchaeota**: (*Methanococcus maripaludis, Methanosarcina acetivoran*)	*Populus trichocarpa*	In vitro assay	^15^N_2_ assay Acetylene reduction assay *nif*H gene amplification	[115]
**Proteobacteria**: (*Azospirillum amazonense* AR3122, *Burkholderia vietnamiensis* AR1122)	Rice (*Oryza sativa* L)	Rice (*Oryza sativa* L.)	Acetylene reduction assay	[116]
**Proteobacteria**: *(Gluconacetobacter diazotrophicus*, *Azospirillum*, *Herbaspirillum seropedicae*, *Herbaspirillum rubrisubalbicans, Burkholderia tropica)*	Sugarcane	Sugarcane	Kjeldahl method. Abundance of ^15^N in leaves Isotopic ^15^N dilution assay	[117]
**Proteobacteria**: (*Pseudomonas* spp., *Rhizobium* spp., *Duganella* spp.)	*Ageratina adenophora*	In vitro assay	Growth on nitrogen-free liquid medium	[118]

**Table 4 plants-10-00935-t004:** List of some examples of stress conditions and the mechanisms of resistance/alleviation by specific bacterial endophytes.

Stress Condition	Bacterial Endophytes Used	Plant Host	Effect/Mechanism of Resistance	References
Drought stress	*Azospirillum* spp.	Maize	- Accumulation of the abscisic acid that regulated plant water balance and osmotic stress tolerance	[78]
Cold tolerance	*Burkholderia phytofrmans* PsJN	Grapevine plant	- Altering the photosynthetic activity and metabolism of carbohydrates involved in cold stress tolerance - The bacterium promoted acclimation and resulted in lower cell damage, higher photosynthetic activity, and accumulation of cold-stress-related metabolites such as starch, proline, and phenolic compounds	[159,160]
Drought stress	*Burkholderia phytofirmans* PsJN, *Enterobacter sp.* PsJN	Maize	- Minimized drought stress - Higher leaf relative water content (30%) - Lower leaf damage in terms of relative membrane permeability - Increasing shoot biomass, root biomass, leaf area, chlorophyll content, photosynthesis, and the photochemical efficiency of PSII	[161]
Drought stress	*Gluconacetobacter diazotrophicus*	Sugarcane (*Saccharum officinarum*)/shoot	- Activation of different genes (ERD15 DREB1A/CBF3 and DREB1B/CBF) - Production of plant hormones (IAA) - Production of proline - Activation of the ABA and ethylene pathways	[162]
Drought stress	*Pseudomonas azotoformans*	*Alyssum serpyllifolium*/leaves	- Improve relative water content - Improve chlorophyll content - Improved oxidative enzyme production (SOD, POD, and CAT) - Improved proline production - Increase plant biomass	[163]
Drought stress	*Bacillus amyloliquifaciens*	Grapevine/roots	- Melatonin secretion - Reduced MDA, H_2_O_2_, and O_2_^−^	[164]
Drought stress	*Pantoea alhagi*	*Alhagi sparsifolia*/ leaves	- Enhancement of the production of IAA, siderophores, soluble sugar, ammonia, EPS, protease enzymes - Decreased the accumulation of MDA - Reduce chlorophyll degradation	[165]
Drought stress	*Bacillus subtilis* and *Paenibacillus illinoinensi*	*Capsicum annuum*/root	- Improvement total biomass, root length, photosynthetic activity, proline contents, transpiration, and cell turgor	[166]
Drought stress	*Bacillus pumilus*	*Glycyrrhiza uralensis*	- Improvement root length - Enhancement of oxidative enzymes (CAT) - Improvement of antioxidant activity (GPX) - Reduced accumulation of MDA, H_2_O_2_, and O_2_^−^	[167]
Drought stress	*Bacillus* sp. strain Acb9, *Providencia* sp. strain Acb11, *Staphylococcus* sp. strain Acb12, *Staphylococcus* sp. strain Acb13 and *Staphylococcus* sp. strain Acb14	*Ananas comosus*	- Enhancement of shoot and root length, and root numbers - IAA production - Nitrogen fixation - ACC-deaminase synthesis. - Active metabolites have antifungal activities	[168]
Drought stress	*Sinorhizobium meliloti*	*Medicago sativa*/root	- Induced the superoxide dismutase (SOD) gene - Upregulation of FeSOD and CU/ZnSOD	[169]
Salinity	*Pseudomonas pseudoalcaligenes*	Rice	- Induced the accumulation of higher concentrations of glycine betaine-like compounds.	[170]
Salinity	*Pseudomonas fluorescens* YsS6 and *P. migulae* 8R6	Tomato plants	- Reduced ethylene levels due to ACC deaminase activity - Higher gain of biomass and a greater number of flowers and buds	[156]
Salinity and trace metals	*Pseudomonas stutzeri* A1501	Rice	- Enhancement of plant growth through secretion of ACC deaminase	[171]
Salinity	*Bacillus* sp., *Pantoea sp.*, *Marinobacterium* sp., *Acinetobacter* sp., *Enterobacter* sp., *Pseudomonas* sp., *Rhizobium* sp. and *Sinorhizobium* sp.	*Psoralea corylifolia* L.	- Enhanced phytohormone production (IAA) - Enhanced seed germination - Enhanced plant vigor index	[172]
Trace metal (copper-contaminated soils)	*Pantoea agglomerans* Jp3-3 and *Achromobacter xylosoxidans* strain Ax 10	*Brassica* sp	- ACC deaminase production - Improved copper uptake by the plants	[173,174]
Salinity	*Bacillus subtilis* strain BERA 71	*Acacia gerrardii* Benth./root	- Inoculated into *Cicer arietinum* seeds and enhanced their growth under saline condition through enhanced plant biomass production; enhanced photosynthetic pigments; reduced ROS levels; enhanced antioxidant enzymes (POD, CAT, glutathione reductase); increased content of total phenols; decreased accumulation of sodium; increased accumulation of N, K, Ca, and Mg; and accumulation of proline	[175]
Salinity	*Curtobacterium oceanosedimentum* strain SAK1, *Curtobacterium luteum* strain SAK2, *Enterobacter ludwigii* strain SAK5, *Bacillus cereus* strain SA1, *Micrococcus yunnanensis* strain SA2, *Enterobacter tabaci* SA3	*Oenothera biennis* L., *Artemisia princeps* Pamp, *Chenopodium* *ficifolium* Smith, and *Echinochloa crusgalli*/roots	- This endophytic bacterial species inoculated into rice plants under salt stress alleviate stress through enhanced phytohormone production (IAA and gibberellins), enhanced organic acid production, reduced ABA content, improved sugar and GSH contents, improved flavin monooxygenase and auxin efflux genes	[176]
Salinity	*Bacillus* spp., *Enterobacter* spp.	*Thymus vulgaris*/leaves, stems, and roots	- Improvement of plant growth through the production of auxins, nitrogen fixation, phosphate solubilization, and extracellular lytic enzyme activities - Reduced antioxidant enzymes in the inoculated plant (tomato) - Secretion active compounds against the plant pathogen (*F. oxysporumen*)	[177]
Trace metals (Cd, Zn, Pb, and Cu)	*Mesorhizobium loti* HZ76 and *Agrobacterium radiobacter* HZ6	*Robinia pseudoacacia/*root nodules	- Enhanced IAA production - Enhanced ACC deaminase - Production of siderophores - Enhanced plant biomass production	[178]
Trace metal (Cd and Ni)	*Enterobacter ludwigii* strain SAK5 and *Exiguobacterium indicum* strain SA22	-	- Enhanced plant growth in presence of trace metal through upregulation of trace metal resistance genes and increased ABA concentrations	[179]
Trace metal (Ni)	*Stenotrophomonas* sp. S20, *Pseudomonas* sp. P21, and *Sphingobium* sp. S42	*Tamarix chinensis*	- Enhanced IAA production, siderophore production, and ACC deaminase	[180]

O_2_^−^, superoxide anion radical; H_2_O_2,_ hydrogen peroxide; MDA, malondialdehyde; SOD, sodium dehydrogenase; POD, peroxidase; CAT, catalase, ABA, abscisic acid; EPS, exopolysaccharide; GPX, glutathione peroxidase; ROS, reactive oxygen species.

**Table 5 plants-10-00935-t005:** Different strategies used by endophytic bacteria to resist phytopathogens.

Strategy Used	Mechanisms	References
Competition for space and nutrients	Competitive root colonization, capacity to stick onto the root, differentiating the growth phase, efficacy to utilize the organic acids existing in the root exudates and hence synthesize different components	[206,207]
Competition with ferric iron	- Siderophore production chelates ferric iron and hence reduces it for pathogen growth.	[196]
Detoxification of virulence factors	- Production of fusaric acids, which detoxify the toxins produced by phytopathogens - Reduced quorum-sensing efficacy through degrading autoinducer signals, hence inhibiting the expression of various virulence genes	[54,208]
Antibiosis	- Production of active compounds such as 2-hexyl-5-propyl resorcinol; pyoluteorin, phenazines, and volatile hydrogen cyanide (HCN)-like compounds; pyrrolnitrin; D-gluconic acid; 6-pentyl-*α*-pyrone; and the volatile 2,3-butanediol - Production of active lipopeptide substances such as iturin, surfactin, polymyxin, fengycin, and bacitracin. - Production of phenols, pyrrolnitrin, phloroglucinol, and volatile organic compounds (VOCs)	[200,209,210,211,212]
Induced systemic resistance (ISR)	- ISR enhanced by the production of pyocyanin, salicylic acid, and siderophores - ISR enhanced via the reaction between chemical elicitors such as chitosan and their derivatives, and endophytic microbes - Production of antioxidant enzymes enhanced ISR	[213,214,215,216]

**Table 6 plants-10-00935-t006:** Some recent examples of bacterial endophyte-mediated biosynthesis of nanoparticles and their activities.

Nanoparticles	Bacterial Endophytes	Plant	Applications	References
Au	*Pseudomonas veronii*	*Annona squamosa*	Antibacterial	[247]
Au	*Pseudomonas fluorescens 417*	*Coffea arabica*	Antibacterial	[248]
CaCl_2_	*Lysinibacillus xylanilyticus*	*Chiliadenus montanus*	Degradation of cellulase	[249]
Cu	*Streptomyces capillispiralis*	*Convolvulus arvensis*	Antibacterial and antifungal	[245]
CuO	*Streptomyces zaomyceticus* and *Streptomyces pseudogriseolus*	*Oxalis corniculata* L.	Antimicrobial, antiphytopathogen, in vitro cytotoxicity, larvicidal activity	[246]
MgO	*Streptomyces coelicolor*	*Ocimum sanctum*	Active against multidrug-resistant microbes	[248]
ZnO	*Sphingobacterium thalpophilum*	*Withania somnifera* (L.)	Antimicrobial	[250]
Ag	*Bacillus siamensis* C1	*Coriandrum sativum*	Antibacterial	[251]
Ag	*Pantoea ananatis*	Monocot plants	Anti-multidrug-resistant	[243]
Ag	*Streptomyces laurentii* R-1	*Achillea fragrantissima*	Antibacterial, in vitro cytotoxicity, and textile industry	[252]
Ag	*Streptomyces antimycotics* L-1	*Mentha longifolia* L.	Antibacterial, in vitro cytotoxicity, and textile industry	[253]
Ag	*Pseudomonas poae* CO	*Allium sativum*	Antifungal	[254]

## Data Availability

The data presented in this study are available on request from the corresponding author.

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
