# Peer review of "Harnessing Bacterial Endophytes for Promotion of Plant Growth and Biotechnological Applications: An Overview"

_plants, 2021, doi:10.3390/plants10050935_

Round 1

Reviewer 1 Report

Title: Looks fine

Abstract: Looks fine

-Maybe the authors should describe first the types of endophytes that exist and then talk about their interaction with the plant.This waySection 3 should be  section 2and section 2 becomes section 3

-Table 1 is not mentioned in the text.Please add it

  • Please put all the tables in the same format. Moreover table 1 itself is not in the same format (for example in the first part you do not separate the different items in columns, letter size etc).The same for the other tables. I do not think it is necessary to add contiunue table 1, continue table 3 etc
  • line 167 replace ornd by and
  • line 174 remove one the
  • line 191 replace some by these
  • line 207-210:: please rewrite it
  • line 215-217: please rewrite it
  • please change title of section 4.1.2 to highlight better the capacity of the endophytes to protect plants against different abiotic and biotic agents

Author Response

Thank you very much for reviewing my manuscript. please see the attachment

Reviewer 2 Report

In this manuscript, the authors reviewed the bacterial endophytes and their potential applications in agriculture and biotechnology. It is a very comprehensive review that focuses on the bacterial endophytes about interaction types and different species. Also, the Authors attempt to summarize applications in the areas of agriculture, bioremediation, production of industrial and pharmaceutical compounds. The authors have covered a large amount of data, and it is difficult to be overly critical.

I have thoroughly read the paper. I think this manuscript covers most of the aspects of bacterial endophytes, but revisions need to be made before considering it for publication. Specifically, language and grammar need significant improvements. Here, providing some comments for the Authors to consider for modification.

  1. Line 30 in Abstract- ammonia is not a fertilizer. Did the Authors mean to say nitrogenous or ammonium?
  2. Line 42- correct the spelling of inhabite
  3. All figures have different fonts and disproportional sizes. Authors need to revise and make it uniform as per Plants Journal format (Arial in the figure?). It will help readers to follow the content. Also, the Authors need to improve the resolution of all figures.
  4. Reference citing format is not uniform, and the Authors need to follow the desired formats for the Plants Bold font for reference number is not required.
  5. Line 74- Rhizobium should be italic.
  6. Why are Tables abruptly divided? I would suggest organizing every table into a single formatting style and make the same font size.
  7. Table 1- last row- Incomplete bracket. Check the whole text for similar mistakes.
  8. Heading and subheadings are confusing. Particularly, the Authors should divide the application part (Section 4) into two parts.

Section 4- Agricultural applications

Section 5- Biotechnological applications

  1. Recent advancements in microbial and plant engineering using modern techniques like CRISPR/Cas are crucial to the agriculture field for crop improvement. Recent papers (for example, https://www.mdpi.com/2076-2607/7/8/269) have summarized the potential of this technology to engineer microbe or plant or both the partners for the desired purpose. Authors may include this aspect in the conclusion section and provide their opinion for future applications in bacterial endophytes.
  2. Line 534- Remove comma at the end of the sentence.
  3. Line 538- Remove “
  4. Reference section- No uniform style is used in reference writing. For example, in some places, a page number is missing (ref. 2), the first letter of each word in the title is capital or not capital?
  5. Reference 16- Plant-Associated <em>Burkholderia</em>??
  6. It seems the Authors are using reference manager software. Still, the Authors should manually check for corrections because of the issues mentioned above.

Author Response

(The authors gave the same response as above.)
